# Insight into the Phylogeny and Binding Ability of WRKY Transcription Factors

**DOI:** 10.3390/ijms23052895

**Published:** 2022-03-07

**Authors:** Kuan-Ting Hsin, Min-Che Hsieh, Yu-Hsuan Lee, Kai-Chun Lin, Yi-Sheng Cheng

**Affiliations:** 1Department of Life Science, College of Life Science, National Taiwan University, Taipei 10617, Taiwan; kt.hish@gmail.com (K.-T.H.); jack3344044@hotmail.com (M.-C.H.); yuslee@ntu.edu.tw (Y.-H.L.); b07b01077@ntu.edu.tw (K.-C.L.); 2Institute of Plant Biology, College of Life Science, National Taiwan University, Taipei 10617, Taiwan; 3Genome and Systems Biology Degree Program, College of Life Science, National Taiwan University, Taipei 10617, Taiwan

**Keywords:** AtWRKY54, phylogenetic tree, flanking region of W-box, binding ability

## Abstract

WRKY transcription factors (TFs), which make up one of the largest families of TFs in the plant kingdom, are key players in modulating gene expression relating to embryogenesis, senescence, pathogen resistance, and abiotic stress responses. However, the phylogeny and grouping of WRKY TFs and how their binding ability is affected by the flanking regions of W-box sequences remain unclear. In this study, we reconstructed the phylogeny of WRKY across the plant kingdom and characterized the DNA-binding profile of *Arabidopsis thaliana* WRKY (WRKY54) based on its W-box recognition sequence. We found that WRKY TFs could be separated into five clades, and that the functional zinc-finger motif at the C-terminal of WRKY appeared after several nucleotide substitutions had occurred at the 3′-end of the zinc-finger region in chlorophytes. In addition, we found that W-box flanking regions affect the binding ability of WRKY54 based on the results of a fluorescence-based electrophoretic mobility shift assay (fEMSA) and quartz crystal microbalance (QCM) analysis. The great abundance of WRKY TFs in plants implicates their involvement in diverse molecular regulatory networks, and the flanking regions of W-box sequences may contribute to their molecular recognition mechanism. This phylogeny and our findings on the molecular recognition mechanism of WRKY TFs should be helpful for further research in this area.

## 1. Introduction

WRKY transcription factors (TFs) are named based on the conserved residue WRKY, which constitutes an integral part of their DNA-binding domain (DBD) and is approximately 60 residues in length. The key structural features of this domain are the conserved DNA-binding motif WRKYGQK and a zinc-finger motif at the C-terminus [1]. Previous research has indicated that WRKY TFs should be classified into three groups based on the number of WRKY domains and the type of zinc-finger motif they contain [2]. The group 1 WRKY TFs contain two WRKY domains, and each of these has a C2H2 motif. The group 2 TFs contain one WRKY domain with a C2H2 zinc-finger motif, whereas the group 3 TFs contain one WRKY domain with a C2HC zinc-finger motif. This grouping of WRKY TFs has been widely used, but molecular evidence has now shown that it is inconsistent with phylogeny [2,3,4]. For example, the *WRKY* genes from *Arabidopsis thaliana* can be grouped into group Ia, comprising *AtWRKY1*, -*2*, *-3*, *-4*, *-10*, *-25*, *-26*, *-32*, *-33*, *-34*, *-44*, and -*58*, and group Ib, comprising *AtWRKY8*, *-12*, *-13*, *-23*, *-24*, *-28*, *-43*, *-45*, *-48*, *-56*, *-57*, *-68*, *-71*, and *-**75*; these two groups are sister groups. Group IIa, comprising *AtWRKY6*, *-9, -18*, *-31*, *-36*, *-40*, *-42*, *-47*, *-60*, *-61*, and *-72*, is followed [4]. However, group Ia and IIa members were treated as a monophyletic group named group IIa by Mohanta et al. [3]. In addition, some members of the groups Ia and Ib that were defined by Wang et al. [4] (*AtWRKY1*, *-2*, *-3*, *-4*, *-10*, *-20*, *-25*, *-26*, *-32*, *-33*, *-34*, *-44*, *-45*, and *-**58*) were designated group I by Eulgem et al. [2]. The remaining members of Wang et al.’s groups Ia and Ib (*AtWRKY8*, *-12*, *-13*, *-23*, *-24*, *-28*, *-43*, *-48*, *-56*, *-68*, *-71*, and *-**75*) were absent from the group I defined by Eulgem et al. [2]. This inconsistency among studies leads not only to improper citations of WRKY TFs, but also confuses postulations about their evolutionary scenario. The confusion may result from the fact that WRKY genes have been classified based on whether they have one or two WRKY domains. [2,3,4]. Furthermore, which type of WRKY gene is closest to the ancestral form remains unclear. Recently, it has been assumed that both types of gene (having one or two WRKY domains) evolved from one ancestral WRKY TF, following diversification and amino acid substitutions in the derived WRKY genes [5]. However, a lack of phylogenetic support makes this proposal debatable.

WRKY TFs are reported to be involved in plant growth, development, metabolism, responses to environmental stresses, and senescence [5]. For example, the WRKY TFs WRKY53, WRKY54, and WRKY70 have been characterized as leaf-senescence regulators [6,7,8]. A line of *Arabidopsis thaliana* overexpressing *WRKY53* showed whole-plant senescence two weeks earlier than the wild type, and a knock-out line showed delayed senescence compared to the wild type [7]. In addition, the expression level of *WRKY53* increased in rosette leaves prior to the expression of several senescence-associated genes (*SAG*), such as *SAG12* [9]. These studies suggest that *WRKY53* is a positive senescence regulator in the senescence process. In contrast, a loss-of-function *wrky70* line showed earlier senescence than wild-type plants, suggesting that *WRKY70* functions as a negative regulator of senescence [8]. *WRKY54*, a homologue of *WRKY70*, displayed similar expression patterns to *WRKY70*, suggesting a redundancy between these two WRKY TFs [6]. *WRKY70* and *WRKY54* are also crucial in the response against pathogenic infection, in that they are involved in regulating the salicylic acid and jasmonic acid signaling that is part of plant defense mechanisms [5,6,10]. Restricted lesion diameter and bacterial growth observed in *wrky54wrky70* double mutants suggest that WRKY54 and WRKY70 serve as negative regulators of the leaf senescence process [6,10,11]. Therefore, WRKY53, WRKY54, and WRKY70 are regarded as three key components in the control of plant senescence and defense processes, which act by receiving external factors and conducting subsequent signaling transduction [5].

To perform their regulatory function in the modulation of gene expression, WRKY TFs bind to a consensus sequence TTGAC-C/T, named W-box, to regulate the expression of target genes [1,12,13]. An extensive survey of binding specificity showed that WRKY TFs exhibit a W-box preference [10]. According to the report, WRKY11 (a member of group IId) binds to the second and eleventh W-box of *Arabidopsis thaliana* senescence-induced receptor-like kinase (*AtSIRK*) promoter, while WRKY26 (group I) binds to the eighth W-box of promoter *AtSIRK* [10]. In addition, they found that WRKY11 binds to W-box with a G residue directly at the 5′ end. In contrast, WRKY26 prefers to interact with W-box with a T, C, or A residue at the same site [10]. As a determinant of the sequence-recognition profile of WRKY TFs, the flanking region of the W-box sequence exerts a profound influence on the DNA-binding preferences of structurally similar members of this highly conserved TF family.

At present, the recognition of specific W-box nucleotide sequence preferences in *A. thaliana* has been elucidated by studying group I (WRKY26, as per Wang et al. [4]) and group II (WRKY11, as per Wang et al. [4]), while the preferences of group III WRKY TFs remains unknown. In the present study, the phylogenetic tree of WRKY across the plant kingdom was reconstructed and the DNA-binding characteristics of WRKY54, a representative of the group III WRKY TFs, were determined. We selected 528 WRKY genes to investigate their origin across the plant kingdom and identify whether the ancestral form of WRKY has single or double WRKY domains. To investigate the DNA-binding preferences of WRKY TFs containing a single WRKY domain, we probed the binding profile of the WRKY54 DBD against a series of W-box DNA sequences by means of a fluorescence-based electrophoretic mobility shift assay (fEMSA) and quartz crystal microbalance (QCM) analysis. In summary, a novel perspective on the evolutionary origin of WRKY TFs is provided through our reconstructed phylogenetic tree. In addition, the binding profile we describe for the WRKY54 DBD sheds light on the long-standing question of the protein–DNA recognition mechanism and the sequence-selective binding of the highly conserved WRKY TFs.

## 2. Results

### 2.1. Retrieval and Identification of WRKY Transcription Factor Genes

WRKY TF gene family was identified by using 16 species from across the plant kingdom (Table 1). These included a rhodophyte, a chlorophyte, embryophytes (two), a tracheophyte, monocots (two), and dicots (eight). In total, we obtained 528 WRKY TF genes. In the selected species, we identified no WRKY TFs in *P. umbilicalis* (rhodophyte), and only two in *Micromonas pusilla* (chlorophyte). Large numbers of WRKY TFs were identified in the monocots (*Oryza sativa*: 75; *Zea mays*: 62) and dicots (*A. thaliana*: 70). We only found WRKY TFs with two WRKY domains in the embryophytes. Based on multiple sequence alignment, there were significant differences in the sequence characteristics of the chlorophyte (*M. pusilla*) WRKY TF and those of most of the other species. Almost all of the WRKY TFs were characterized by a conserved WRKY motif (xWRKYGQK or xWRKYGEK) with a zinc-finger motif (CxCxHTC or CxCxHxH), where x denotes any amino acid. However, the WRKY TF from *M. pusilla* (Mpu50253) contained a conserved WRKY motif (RWRKYGQK) with only part of the zinc-finger motif (C), suggesting that three or four conserved amino acid substitutions (C → CxCxHxH or C → CxCxHTC) occurred after plants had colonized land.

### 2.2. Phylogenetic Reconstruction for WRKY Transcription Factors

To trace the evolution of WRKY TFs, the WRKY TF obtained from *M. pusilla* (Mpu50253, no zinc-finger motif) was chosen as the root of the reconstructed WRKY phylogeny (Figure 1). In the resulting phylogeny, the TFs were grouped into five clades: Clade 1 (BI: 1; ML: 1), Clade 2 (BI: 0.99; ML: 0.96), Clade 3 (BI: 0.87; ML: 0.8), Clade 4 (BI: 1; ML: 0.9), and Clade 5 (BI: 0.8; ML: 0.9) (Figure 1).

Clade 1 comprised the WRKY TFs obtained from *O. sativa*, *A. thaliana*, *S. lycopersicum*, *P. persica*, *Fragaria* x *ananassa*, *M. acuminata*, *A. trichopoda* and *H. annuus*. Three types of zinc-finger motifs were identified in this clade: CxCxHNH (SlWRKY15, AtWRKY55, and OsWRKY57), CxCxHTC (Ha94651) and CxCxHRH (OsWRKY68 and WRKY17).

Clade 2 contained WRKY TFs from across the plant kingdom, from *Physcomitrium patens* (e.g., Pp3000) to *Prunus persica* (e.g., Ppe74300). There were two main zinc-finger structure types identified in Clade 2: CxCxHNH (e.g., WRKY71, Mp6s0129, Spol02923, and Aco99600) and CxCxHTH (e.g., WRKY48, Zm07329, OsWRKY11, and Spol04568). A WRKY TF lacking a zinc-finger motif, OsWRKY52, was also located in Clade 2.

In Clade 3, the major zinc-finger motif types were CxCxHNH (e.g., Ha38981, WRKY9, and F215A8), CxCxHTH (e.g., Pp15040 and Pp32160) and CxCxHTC (e.g., WRKY38, SlWRKY55 and WRKY46). Notably, no WRKY TFs from the monocots was included in Clade 3.

In Clade 4, the WRKY TFs containing two WRKY domains were dominant; these were found in embryophytes, monocots, and dicots (purple branches in Figure 1). The predominant zinc-finger motif was CxCxHNH-CxCxHNH (e.g., Pp23590, OsWRKY78, and WRKY1). The zinc-finger motif in the WRKY TFs with only one WRKY domain was CxCxHNH (e.g., WRKY10).

Lastly, Clade 5 comprised only TFs with a single WRKY domain and a single zinc-finger motif, and these were assigned to monocot (brown-yellow branches in Figure 1) and dicot subclades (pale blue branches in Figure 1). Within the Clade, the main zinc-finger motif types were CxCxHNH (e.g., Zm39532, Spol03061, SlWRKY27, and Ha36511), CxCxHSH (e.g., OsWRKY5) and CxCxHTC (e.g., OsWRKY20 and Zm23616). There were also WRKY TFs without a complete zinc-finger motif (e.g., WRKY18) in the Clade 5.

Finally, the long branches observed in the reconstructed WRKY phylogeny suggest a high degree of nucleotide variation across the plant kingdom.

### 2.3. The Single WRKY Domain of WRKY54 Exists as Both a Monomer and in an Aggregated Form In Vitro

To study the binding ability of the single WRKY domain in Clade 3, the recombinant WRKY54 protein was expressed and purified. In the first trial, the full length of the WRKY54 recombinant protein exhibited an aggregated form with no DNA binding ability in solution after cell lysis. The DNA binding domain of WRKY54 from residues 133–224 was constructed. After protein expression and purification, the recombinant WRKY54 DBD protein exhibited two forms—an aggregated form and a monomeric form—that corresponded to two major peaks in the SEC elution profile (Figure 2A). SDS-PAGE showed that main protein product size of the eluted solution collected at 43 mL, 78 mL, and 81 mL was 50–75 kDa (Figure 2B). To verify the oligomerization state of the WRKY54 DBD, protein solutions collected at 43, 78 and 81 mL were subjected to DLS. The protein solutions collected at 78 and 81 mL may have contained the same protein product, so these solutions were pooled for the DLS analysis. The DLS results showed that the molecular mass of the WRKY54 DBD fraction collected at 43 mL was 1667 kDa, suggesting that the WRKY54 DBD in this fraction was in the aggregate form (Figure 2C). In contrast, the protein size measured in the pooled solutions collected at 78 and 81 mL was 59 kDa, suggesting that the WRKY54 DBD in these fractions was in the monomer form (Figure 2D).

### 2.4. WRKY54 DNA-Binding Domain Can Bind to W4 Box from SAG12 Upstream Sequence

It is known that WRKY TFs bind to the DNA sequence named W-box (5′-TTGAC-C/T-3′). However, several sequences upstream of the promoters triggered by WRKY TFs contain W-box–centered sequences with TTGACC or TTGACT. Therefore, we first used a fEMSA to examine whether the WRKY54 DBD can bind to W-box sequences with TTGACC or TTGACT from the *SAG12* upstream region. Four types of W-box (W1–W4) obtained from *SAG12* upstream regions were used as probes (Figure 3A). Those four types of W-box were labeled with fluorescein and incubated with purified WRKY54 DBD protein. The results confirmed binding of the WRKY54 DBD protein to W4, but no binding to W1, W2, or W3 (Figure 3B). These four W-box sequences differ in the nucleotides at either the 5′ or the 3′ flanking region (Figure 3A). To examine whether the nucleotides located adjacent to W4 box affect the binding ability of WRKY54, we designed a series of W4 probes.

### 2.5. WRKY54 Can Bind to the W4 of SAG12 In Vivo

The ChIP-PCR technique was used to confirm whether WRKY54 binds specifically to W4 in vivo. Consistent with our fEMSA results, WRKY54 failed to bind to W1–W3 in either wild type *Arabidopsis* or a WRKY54-overexpression line (Figure 4A–C). However, for W4, a clear banding pattern could be seen for the WRKY54-overexpression line, although it was weaker for the wild type (Figure 4D).

### 2.6. Length of the Flanking Region Adjacent to W-Box Affects the Binding Ability of WRKY54 DBD

We then examined whether the composition of the flanking region adjacent to W4 box affected the specific binding ability of the WRKY54 DBD. Six artificial W4 variants were synthesized for this purpose (Figure 5A). Clear banding shift patterns were observed for the WRKY54 DBD–W4, WRKY54 DBD–T13, and WRKY54 DBD–T12a pairs (Figure 5B). In contrast, the WRKY54 DBD–T12 pair displayed a relatively weak banding shift. For the pairs involving T6, T8, and T10, no banding shift was observed, suggesting that the WRKY54 DBD was unable to bind to those artificial W4-like nucleotides (Figure 5B).

The QCM was then used to examine the binding constants of the WRKY54 DBD to W4 and the W4-like nucleotides (Table 2). The *K*_d_ value obtained from the W4–WRKY54 DBD pair was 163 ± 10.06 pM. The *K*_d_ values were 192 ± 22.5 pM and 2195 ± 442.3 pM for the T13–WRKY54 DBD and T12–WRKY54 DBD pairs, respectively. When the 5′-end nucleotides were removed, the *K*_d_ values were 47 ± 6.2 pM (T12a–WRKY54 DBD pair) and 68 ± 11.73 pM (T13–WRKY54 DBD pair; Table 1, Figure 5C). There were no signals detected in the T6, T8, or T10 pairs. These results showed that a flanking region with at least three nucleotides at the 5′ end of TTGACT is required for the binding of the WRKY54 DBD.

### 2.7. Structural Insights into the DNA Binding of WRKY54 to W-Box

The three-dimensional model of the docked structure of the WRKY54 DBD revealed a four-stranded antiparallel β-sheet with the conserved WRKYGQK (W157, R158, K159, Y160, G161, Q162, and K163) motif located in the β1 strand. Insertion of the four-stranded antiparallel β-sheet into the major groove of W-box DNA permits extensive interactions between the residues of the WRKY54 DBD and the DNA nucleotides (5′-ATTTGTTGACTAGG-3′, W-box sequence underlined), as illustrated in Figure 6A. At the protein–DNA interface, intermolecular contacts between the WRKY54 DBD and the W4 nucleotides are primarily between the conserved residues W157, R158, K159, Y160, G161, and K163 on the β1 strand and dT4–dT7 and dG19–dA23 on the DNA. In this context, molecular interactions are mediated mainly by the formation of apolar contacts and H-bonds between the conserved residues of the WRKYRQK binding motif (W157, R158, K159, and Y160) and the DNA nucleotides dG5, dT6, dG19, dT20, dC21 and dA22 (Figure 6B). In this postulated scenario, intermolecular hydrogen-bonding interactions formed between W157 and dT4 and/or dG5; K159 and dT6; and K163 and dG19 and/or dT20 are involved in the specific binding of the WRKY54 DBD to the W-box DNA sequence (Figure 6C,D). Based on the binding mode of the WRKY54 DBD to W4, a 5′ flanking region of W-box with at least three additional nucleotides is thus required for WRKY54 binding.

## 3. Discussion

### 3.1. WRKY TFs Can Be Classified into Five Clades

WRKY TFs are involved in metabolism, growth, pathogen resistance, and abiotic stress responses [5]. The phylogeny of WRKY TFs has therefore been reconstructed several times to identify the WRKY groups and their evolutionary process [2,3,4]. Surprisingly, these previous studies were unable to create a consensus on the evolution of WRKY TFs [2,3,4]. These TFs contain two types of factors with highly diverse sequences and gene structures, which may explain the inconsistent topologies that were derived in these previous studies [3]. Here, we conducted a new phylogenetic analysis of WRKY TFs using data from an up-to-date dataset (16 representative species selected from across the plant kingdom) to elucidate the relationships between these WRKY TFs. By setting the WRKY obtained from *M. pusilla* (Mpu50253) as the root, we obtained five distinct clades with moderate (BI: 0.88; ML: 0.8) to strong (BI: 1; ML: 1) branch support (Figure 1). An evolutionary history of the WRKY domain can therefore be proposed based on the amino acid sequences next to the schematic WRKY phylogeny in Figure 7.

In this study, WRKY TFs with the sequence RWRKYGK…RSYYKC in the Chlorophyta were found, but not in the Rhodophyta (Figure 7A, Table 1). This suggests that these WRKY TFs may have originated in Chlorophyta and then evolved in land plants. The WRKY domain is currently regarded as a sequence that exists in all land plants, is approximately 60 amino acid residues long, and contains a WRKY DBD and a zinc-finger motif (CxCxHNH or CxCxHTC) [2,5,7]. However, the incomplete zinc-finger motif identified in *M. pusilla* (Mpu50253) suggests that the currently known WRKY domain with a zinc-finger motif only appeared in land plants after the development of chlorophytes (Figure 7A). After the zinc-finger motif had developed, a series of amino-acid substitutions and duplications occurred that follow WRKY TF divergence.

By mapping main WRKY motif and zinc-finger motif next to simplified WRKY topology, a potential WRKY evolution pattern was revealed in plants (Figure 7). After Chlorophytes, land plants gained three types of zinc-finger structures, including CxCxHNH, and CxCxHTC (Figure 7B). We regarded these as the basal forms for WRKY TFs. Following divergence, a basal form, CxCxHNH, was dominant within Clade2. In addition, amino acid residue substitutions at 3′ end of zinc-finger leading to a second zinc-finger type –CxCxHxH in Clade2. For example, a threonine replacement at zinc-finger 3′end formed a CxCxHTH zinc-finger structure in *A. thaliana* WRKY48 or a valine replacement at zinc-finger 3′end formed a CxCxHVH in *M. acuminata* Mu7T13310. Lastly, loss of zinc-finger structure was found in Clade 2, like OsWRKY52. This could due to amino acid similarity between conserved WRKY region of OsWRKY52 and other Clade 2 WRKY sequences. Next, CxCxHNH, CxCxHTH and CxCxHTC zinc-finger structure were found in Clade 3. The WRKY TFs containing two WRKY domains were assigned to Clade 4, suggesting a post-species divergence WRKY-domain-duplication event. In Clade5, two main zinc-finger structures were found, including CxCxHNH and CxCxHTC. One of the common features of Clade 5 is the CxCxHTC zinc-finger structure (Figure 7A), suggesting zinc-finger motif ancestral polymorphism maintenance or an amino-acid substitution occurring in their common ancestor. Substitutions in zinc-finger structure may associate with diverse functions of WRKY TFs in plants. That is because zinc-finger structure of WRKY TFs plays a key role in mediating WRKY protein binding to W-box nucleotide ability and also involving in oligomerization state transition of WRKY TFs [11,12]. Based on these phylogenetic relationships and the zinc-finger structures, the WRKY domain obtained the full zinc-finger motif after the emergence of chlorophytes, after which amino-acid substitutions occurred in the zinc-finger motif, followed by divergence that led to WRKY domain duplication in the common ancestor of Clade 4 and a retained ancestral zinc-finger motif polymorphism in Clade 5 (Figure 7B). Previous models have inferred WRKY evolution based mainly on nucleotide similarity, suggesting that the current WRKY TFs evolved from TFs containing a double WRKY domain [5,13]. Unlike these previous models, our evolutionary scenario for the current WRKY TFs seen across the plant kingdom was based on development from a TF containing a single WRKY domain.

### 3.2. Molecular Binding Mechanism for Specific WRKY–W-Box Interaction

Based on the binding profile of the WRKY54 DBD to W1–W4 sequences, the DNA binding ability is abolished when probed with W-box (TTGACT) alone (Figure 3 and Figure 4) in fEMSA assay. In contrast, the WRKY54 DBD could be bound well to W4 with 5′-TTG and 3′-A residues immediately adjacent to the W-box (W4) nucleotides (Figure 5, Table 1). Based on these results, we identified the W-box flanking region as an essential element in WRKY54 DBD binding interactions [8,10]. The relative importance of the W-box flanking region has been implied in previous research on the DNA binding ability of other WRKY TFs (e.g., WRKY11, WRKY6, WRKY26, and WRKY38) [10]. For instance, it is known that WRKY11 and WRKY6 bind to W-box sequences that have a G residue directly adjacent to the 5′ end of the conserved W-box nucleotides, whereas WRKY26 and WRKY38 bind to conserved W-box nucleotides with a T, C, or A residue in the same place [10]. Combined, these findings suggest that the composition of the W-box flanking regions affect the ability of these W-box nucleotides and WRKY TFs to bind [10].

In addition, the present study provides a novel clue to the DNA-binding mechanism of WRKY TFs. As suggested by the fEMSA assay and QCM analysis, the minimum length necessary to maintain the specific interaction between W4 and the WRKY54 DBD is three retained nucleotides at either end of the conserved region of W4, suggesting that the length of W-box flanking regions could be a factor in the DNA binding of the WRKY54 DBD (Figure 5, Table 1).

The DNA binding mode of the WRKY54 DBD was proposed based on our structural model with W4 DNA. Consistent with the binding mode of the WRKY1 domain (PDB ID: 6J4E) [14], which was the template structure we used for our homology modeling of the WRKY54 DBD, the protein–DNA binding interaction was primarily mediated by the conserved WRKYGQK binding motif. However, we observed a slight difference in the structural arrangement of the loop between β1 and β2, and of another loop connecting β3 and β4 at the C-terminus. As illustrated in a previous study [15], the dynamic process of WRKY TF binding to W-box DNA involves structural rearrangements of the β1 strand harboring the conserved DNA-interacting residues. Accordingly, it has been proposed that this structural change may alter the loop structure and the length of the strands, enabling extensive contact with the DNA [15]. On the other hand, because of the highly similar DNA-binding mode used by WRKY TFs from different groups [14,15], the mechanism underlying the DNA-binding preferences of the WRKY TFs remains elusive. Despite the limited evidence, it has been noted that differences in the DNA-contacting residues among WRKY TFs may be the key factor determining their preferences for different W-box flanking regions. For example, Yamasaki et al. [15] suggested that the hydrogen-bonding interaction between R415 and dG5 in the modeled structure of WRKY4 binding to W-box (5′-CGCCTTTGACCAGCGC-3′) may be responsible for recognizing the nucleotides in the W-box flanking region, which could explain the binding preferences of WRKY TFs. Interestingly, in structure of WRKY54 DBD binding to W4 DNA, the dT4 located adjacent to the 5′ end of the W-box core region could form a hydrogen bond with the conserved residue W157. In accordance with our fEMSA results, the 5′-TTG located adjacent to the W4 box nucleotides was shown to be indispensable for the W4–WRKY54 DBD interaction. Based on these results, we propose a “recognition-then-binding” mechanism for the formation of the W4–WRKY54 DBD protein–DNA complex.

Based on our results and previous studies, the interaction between WRKY TFs and W-box nucleotides seems to occur in a clade-specific manner [10,15]. For example, the 5′-TTG of W4 is essential for the protein–DNA interaction between WRKY54 and W4 (Figure 5). However, the 5′-AG nucleotides of W2 of PR1-1 (pathogenesis-related protein 1-1) and the 5′-T of the conserved PR1-1 W2 are necessary for the protein–DNA interaction between WRKY11 and W2 of PR1-1 [10]. Further, the protein–DNA interaction between WRKY26 and W2 of PR1-1 is mainly affected by mutations of both the 5′- and 3′-end nucleotides of the conserved W2 region [10]. In reconstructed phylogeny, WRKY11, WRKY54, and WRKY26 belong to Clade 1, Clade 3, and Clade 4, respectively (Figure 1), suggesting the existence of clade-specific interactions between WRKY TFs and their corresponding W-box sequences.

Here, an association between the recognition specificity of WRKY TFs with their corresponding W-box sequences and their phylogeny was established. However, one other interesting issue emerged: how a particular WRKY TF is involved in various molecular regulatory networks. WRKY TFs are known to regulate complicated developmental processes (e.g., leaf senescence) and responses to environmental cues (e.g., responses to pathogen infection) [5,6,15,16,17,18,19,20,21,22]. For example, WRKY26 is known to regulate *PR1* (a pathogenesis-related protein in *Arabidopsis*) and *SIRK* (a senescence-associated expression gene in *Arabidopsis*) by identifying different W-box nucleotides (PR1 W2: AGTTGACCAA; SIRKW11/W12: TTGGTTGACTATCAACATCTTATTGACCAAAT) [10,21]. To be able to interact with such different W-box sequences, a conformational change in WRKY26 at the protein level might be expected. Further analyses of the protein–DNA interactions of specific WRKY domains involved in different molecular regulatory networks may provide clues that can lead to a full understanding of the mechanisms of plant-specific WRKY TFs. In summary, WRKY TFs can recognize the flanking regions of W-box sequences to enhance their binding ability and ensure transcription regulation.

## 4. Materials and Methods

### 4.1. WRKY Phylogeny Reconstruction

First, the core residues of WRKY domains (53–55 residues in length, on average) of green plants were labeled using Bioedit to identify conserved *WRKY* genes [23]. Then, to investigate *WRKY* gene evolution across the plant kingdom, *WRKY* genes were retrieved from the online Phytozome database (https://phytozome.jgi.doe.gov/pz/portal.html# accessed on 23 January 2022) [24] and the database of the National Center for Biotechnology Information (NCBI). The 73 *WRKY* genes of *Arabidopsis thaliana* (*AtWRKY*) were used as templates for identifying the *WRKY* genes of the following selected species: *Micromonas pusilla*, *Marchantia polymorpha*, *Selaginella moellendorffii*, *Amborella trichopoda*, basal angiosperms (such as *Amborella trichopoda*), monocots (including *Zea mays* and *Oryza sativa*), and eudicots (including *Aquilegia coerulea, Solanum lycopersicum* and *Arabidopsis thaliana*) (for details see Table 1 and Appendix A). In addition, two polyploidy species are also included in this study, such as *Musa acuminate* (triploid) and *Fragaria* x *ananassa* (octoploid). These species are chosen to represent major plant lineages. Although we searched the *Porphyra umbilicalis* (Rhodophyta) database, implemented in Phytozome, we found no WRKY-containing genes in that database. Also, a *WRKY* gene found in *M. pusilla*, Mpu62993, caused long-branch attraction in preliminary phylogenetic reconstruction. Thus, Mpu62993 was excluded from the reconstruction. To conduct a comprehensive search for the *WRKY* genes of the selected species, the default threshold value and tblastx were used for each search. Only sequences containing a WRKY domain were used for the phylogenetic reconstruction.

First, all *WRKY* nucleotide sequences were aligned using the MUSCLE algorithm [25] in MEGA v.6 [26]. Next, the resulting alignment matrix was visually refined based on amino acid translations using Bioedit [23]. The best-fitting model of the aligned WRKY matrix was GTR + G + I, as evaluated using the Bayesian information criterion (BIC) [27]. Both Bayesian inference (BI) and maximum likelihood (ML) algorithms were applied to infer the relationships among the identified WRKY genes using the PhyML 3.0 online interface [28]. Branch support for nodes was assessed using an approximate likelihood ratio test (aLRT) [29] and a Bayesian-like transformation of the aLRT (aBayes) [30] for the ML and BI findings, respectively (see Appendix A for details). A conventional bootstrap algorithm was conducted with 1000 replicates. The reconstructed phylogeny was visualized using the online interface Interactive Tree of Life (iTOL) v5 [31].

### 4.2. Vector Construction for the WRKY54 DNA-Binding Domain

For protein expression and purification, the wild type *WRKY54* gene (AT2G40750) was amplified via polymerase chain reaction (PCR) from the plasmid pET21a-*WRKY54*, which was provided by Dr. Shih-Tong Jeng (Institute of Plant Biology, National Taiwan University, Taipei, Taiwan). The DBD of *WRKY54* (WRKY54 DBD) was amplified with the forward *Bam*H1 primer (5′-CCCGGATCCGGATGCTACACTAGAA-3′) and the reverse *Pst*1 primer (5′-CCCCTGCAGGAAAAGGCTCGGTCTT-3′). It was then subcloned into the expression vector pMALH12-c5v, such that the recombinant protein WRKY54 contained a maltose binding protein tag, a thrombin cleavage site at the N-terminus, and a twelve-histidine tag at the C-terminus.

### 4.3. Protein Expression and Purification

The WRKY54 DBD recombinant protein was expressed using *Escherichia coli* Rosetta-gamiB (DE3). Cultures were incubated at 37 °C to an OD_600_ of 0.4–0.6 and induced by adding 0.1 mM isopropyl β-D-1-thiogalactopyranoside (IPTG), then grown overnight at 16 °C. Bacterial cells were pelleted and lysed by sonication in 25 mL of lysis buffer (30 mM 4-(2-hydroxyethyl)-1-piperazineethanesulfonic acid [HEPES], pH 7.5, 0.5 M NaCl, 20 mM imidazole) to obtain the WRKY54 DBD recombinant protein. After cell lysis, the cell debris were centrifuged for 25 min at 12,500× *g* (J2-MC, Beckman, IN, USA). The supernatants were filtered through a 0.45 μm filter (Millipore, Burlington, MA, USA) and loaded into a 5 mL Ni^2+^-Sepharose resin column (HisTrap FF, GE Healthcare, Chicago, IL, USA). The column was washed with 10 times the column volume of binding buffer prior to eluting the WRKY54 DBD recombinant protein. After the column washing, 50 times the column volume of elution buffer, complemented with 40 mM imidazole, was injected to elute the recombinant protein. The molecular mass of the purified 12 × His-WRKY54 DBD was estimated as 59 kDa based on sodium dodecyl sulphate–polyacrylamide gel electrophoresis (SDS-PAGE) (Figure 1).

For advanced purification, all proteins were passed through a Superdex S-75 column (GE Healthcare) with elution buffer (30 mM HEPES, pH 7.5, 0.5 M NaCl, 500 mM imidazole). The molecular mass of the WRKY54 recombinant protein was estimated using a Superdex S-200 column and calibrated with gel filtration standard protein markers (BioRad, Hercules, CA, USA: thyroglobulin, 670 kDa; γ-globulin [bovine], 158 kDa; ovalbumin [chicken], 44 kDa; myoglobin [horse], 17 kDa; vitamin B12, 1.35 kDa). After size-exclusion chromatography (SEC), all proteins were concentrated using Amicon Ultra-15 centrifugal filters (Merck, Darmstadt, Germany) and quantified using a DS-11 spectrophotometer (DeNovix, Wilmington, DE, USA).

### 4.4. Determination of Molecular Size via Dynamic Light Scattering

The protein samples were subjected to dynamic light scattering (DLS) after advanced purification by SEC. Data were collected using a Zetasizer Nano ZS DLS instrument (Malvern Instruments, Malvern, UK) equipped with 50 mW laser fiber. An appropriate refractive index, viscosity (10% glycerol), and temperature (25 °C) was set for each sample.

### 4.5. DNA Binding Assays via fEMSA

To visualize whether the WRKY54 DBD interacted with specific W-box regions of the *SAG12* gene, a series of fEMSA assays were conducted. All W-box nucleotides (including W1 to W4, which were identified from the *SAG12* promoter region, and artificial synthesized W-box–like nucleotides) were labeled with fluorescein at the 5′ end and used as DNA probes in the fEMSA. Prior to native polyacrylamide gel electrophoresis, the DNA probe–protein mix was incubated in the dark for 30 min. The DNA–protein binding reaction was performed by incubating purified WRKY54 DBD recombinant protein with the aforementioned double-stranded W-box and W-box–like nucleotides at 4 °C for 90 min at 120 V. After electrophoresis, the DNA probe–protein binding pattern was observed using a fluorescent luminescence image analyzer (FluorChem M, San Jose, CA, USA) at National Taiwan University.

### 4.6. Determination of Binding Constant between WRKY54 DBD Protein and W4 or W4-like DNA Using Quartz Crystal Microbalance Technique

The QCM technique was used to determine the dissociation constant (*K*_d_) values and examine the binding ability of the WRKY54 DBD protein with W4 DNA or W4-like DNA. The protein–nucleotide interactions in WRKY54 DBD–W4 DNA and –W4-like DNA pairs were analyzed by using an AffinixQN QCM biosensor (Initium, Tokyo, Japan). To measure *K*_d_, one has to describe the relationship between pressure and the number of active sites on the surface undergoing adsorption by applying the Langmuir equation ([32]):$$\left[\mathrm{Host}\right]+\left[\mathrm{Guest}\right]\begin{array}{c} Kon\\ \leftarrow \\ \to \\ Koff \end{array}\left[\mathrm{Host}\right]\left[\mathrm{Guest}\right]$$

Here, [Host] represents the adsorbate surface and [Guest] represents the adsorbate molecule, while [Host][Guest] indicates that the [Host] and the [Guest] have formed an adsorbed complex. *K*_on_ and *K*_off_ represent the association and dissociation rate constants, respectively. The ratio of *K*_off_ to *K*_on_ is *K*_d_, the dissociation constant.

Prior to usage, the QCM biosensor was washed twice with 3 μL of piranha solution (H_2_SO_4_ and H_2_O_2_ in a 3:1 ratio) and incubated with 1% SDS for 5 min. Then, 400 μL of reaction buffer (30 mM HEPES, pH 7.5, 0.5 M NaCl, and 2 mM tris(2-carboxyethyl) phosphine [TCEP]) was applied to the dried sensor to balance and set up the magnetic stir at 1000 rpm at 25 °C. Avidin (1 mg/mL) was injected into the reaction buffer and coated on the Au electrode plate until saturation. Excess proteins were washed out with the reaction buffer until the oscillation frequency became a static horizontal line. Next, 5 μL of a double-stranded DNA probe with a biotin-labeled 5′-end (50 μM) was added. When the frequency had stabilized at ca. 0.3 Hz/s, 1 mg/mL of WRKY54 DBD recombinant protein was injected every 10 min, 10 times. The continuous titration method was used to determine the binding constant of the protein–DNA pairs, and the values were recorded as multiple binding curves using the AffinixQN v2 software (Initium, Tokyo, Japan). Data from three independent repeats were processed using AQUA v2 software (Initium, Tokyo, Japan).

### 4.7. Examination of WRKY54 Protein and W-Box through Chromatin Immunoprecipitation In Vivo

The chromatin immunoprecipitation (ChIP) technique was used to examine whether WRKY54 interacts with the W-box identified from *SAG12* gene in vivo. Chromatin was extracted from 5.5-week-old *Arabidopsis* plants. After fixation with 0.37% formaldehyde, the chromatin was sheared to an average length of 500–1000 bp by sonication and then immunoprecipitated with rabbit-anti-myc antibodies (Abcam, Cambridge, UK, ab9106). The cross-linking was then reversed, and the quantity of each precipitated DNA fragment was determined via PCR using specific primers (Appendix A). Three biological replicates were performed.

### 4.8. Homology Modeling and Protein–DNA Complex Docking

To gain insight into the probable binding mode of WRKY54, homology modeling of the WRKY54 DBD was performed in PyMod 3 [33] with the crystal structure of the *WRKY1* domain (PDB ID: 6J4E) as a template model. Next, protein–DNA docking was carried out using the ZDOCK server [34]. From the 10 top-scoring docking poses generated by the rigid-body docking algorithm, a final model was selected based on information provided in previous protein–DNA binding studies [14].

## 5. Conclusions

WRKY transcription factors (TFs) participate various molecular regulatory networks in plants. In this study, a reconstructed WRKY TFs phylogeny shows that WRKY TFs may have originated in Chlorophyta and then diversified with amino acid substitutions at zinc finger motif in land plants. In addition, minimum length and specific nucleotide of W-box flanking region are two key factors in affecting binding and recognition ability of WRKY TFs based on the results obtaining from studying binding and recognition ability of WRKY54 DBD and its corresponding W-box nucleotides.

## Figures and Tables

**Figure 1 ijms-23-02895-f001:**
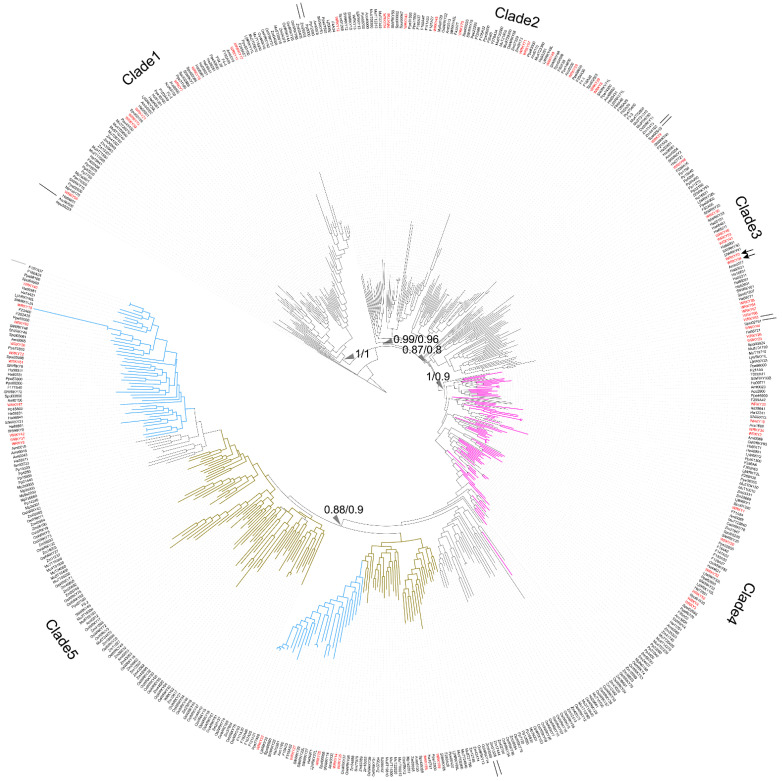
Phylogeny of WRKY transcription factors (TFs) across the plant kingdom. Red names: WRKY TFs obtained from *A. thaliana*. Purple branches: WRKY TFs containing two WRKY domains. Pale blue branches: the dicot WRKY TF subclade. Brown-yellow branches: the monocot WRKY TF subclade. Arrows: WRKY TFs related to the leaf-senescence process in *A. thaliana*, including WRKY54 and WRKY70. Triangles indicate Bayesian inference/maximum likelihood supporting statistic values respectively.

**Figure 2 ijms-23-02895-f002:**
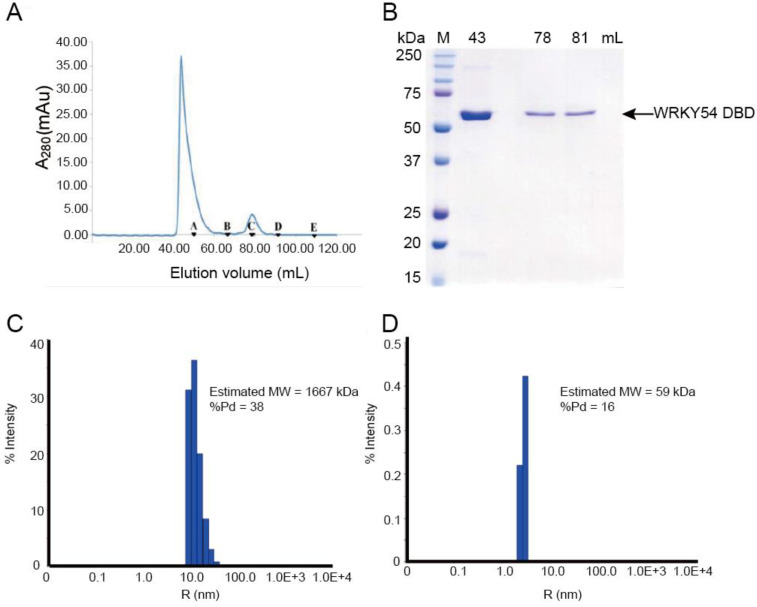
The WRKY54 DNA-binding domain (DBD) tends to form both aggregated and monomeric states in vitro. (**A**) Size-exclusion chromatography (SEC) analysis reveals that the WRKY54 DBD could be eluted as either an aggregated or a monomeric form in elution buffer (30 mM HEPES, 0.5 M NaCl, pH 7.5). Black triangles denote the protein standards: A, thyroglobulin (670 kDa); B, γ-globulin (158 kDa); C, ovalbumin (44 kDa); D, myoglobin (17 kDa); E, vitamin B_12_ (1.35 kDa). (**B**) SDS-PAGE analysis of eluted WRKY54 DBD recombinant protein. M, protein markers; 43, 78, and 81 mL represent the peaks of WRKY54 protein elution in the SEC analysis. (**C**) Dynamic light scattering results revealed the aggregated form of the WRKY54 DBD protein at an elution volume of 43 mL (molecular weight [MW]: 1667 kDa). (**D**) Dynamic light scattering results revealed the monomeric form of the WRKY54 DBD protein at an elution volume of 78–81 mL (MW: 59 kDa).

**Figure 3 ijms-23-02895-f003:**
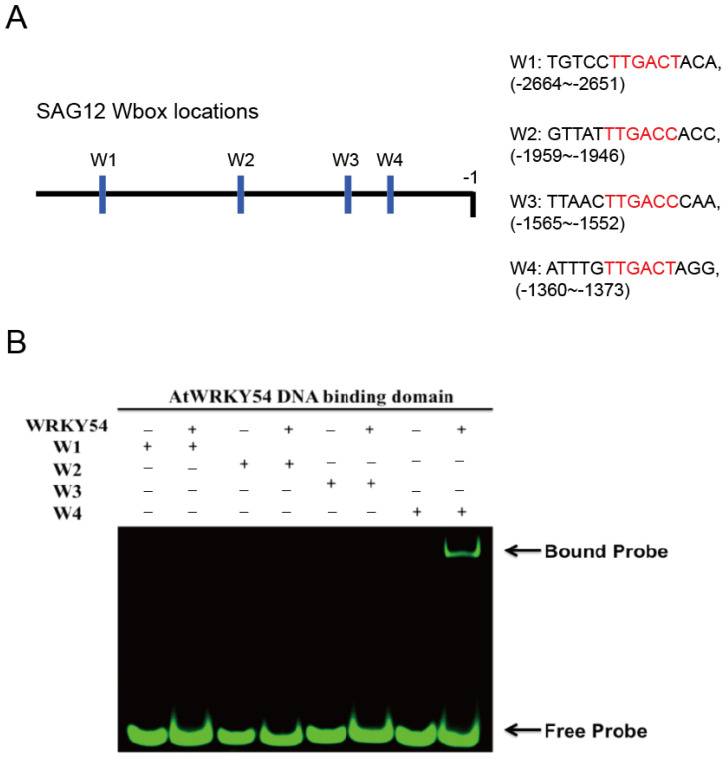
Binding preference of WRKY54 DNA binding domain (DBD), as revealed by fluorescence-based electrophoretic mobility shift assay (fEMSA). (**A**) Location and nucleotide composition of the four identified W-box regions adjacent to the *SAG12* gene. Conserved W-box regions are labelled in red. (**B**) W1–W4 correspond to the four W-box regions shown in (**A**). The fEMSA of the four fluorescence-labeled W-box regions incubated with purified recombinant WRKY54 DBD protein.

**Figure 4 ijms-23-02895-f004:**
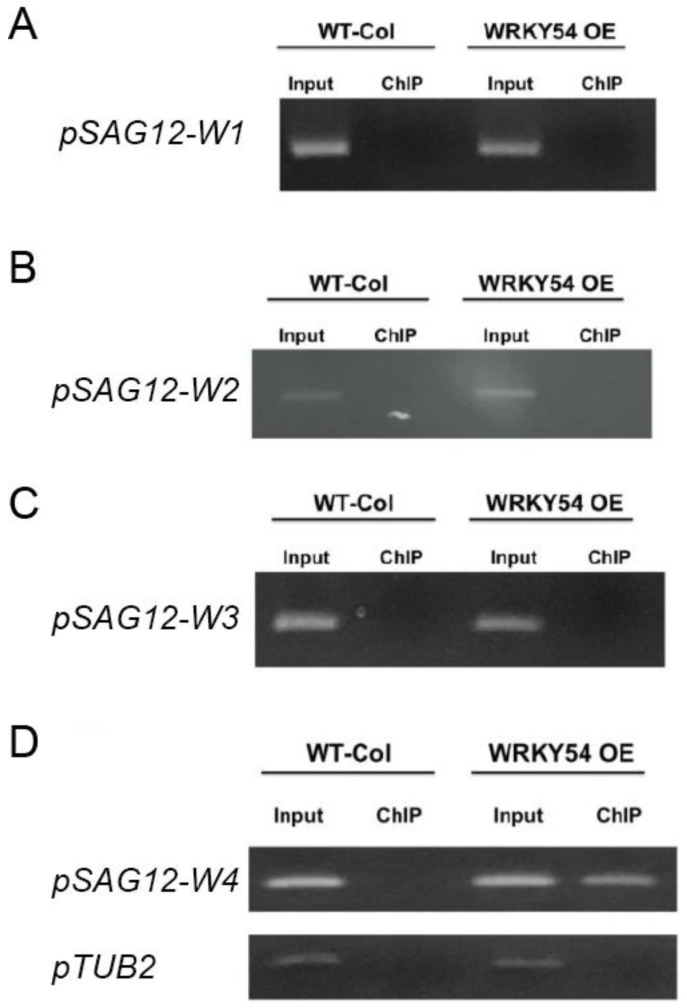
Specific binding ability of WRKY54 to the four W-box types in *SAG12* in vivo. (**A**–**D**) *pSAG12-W1* to *pSAG12-W4* represent the W1–W4 regions shown in Figure 3. WT-col: wild-type *Arabidopsis*. WRKY54 OE: WRKY54-overexpression line. Input: PCR products amplified from genomic DNA. ChIP: PCR products amplified from rabbit-anti-myc precipitation DNA. Left panels represent the ChIP-PCR verification of the relevant W-box type from *A. thaliana* Columbia-0. Right panels represent ChIP-PCR verification of the relevant W-box type from the WRKY54 OE line.

**Figure 5 ijms-23-02895-f005:**
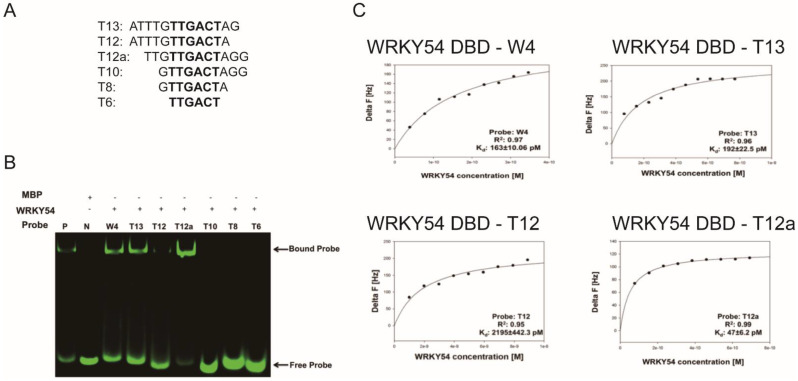
Length preference of WRKY54 DNA-binding domain (DBD) for W4 variants. (**A**) Artificially truncated flanking region of the W4 box sequences used to identify the nucleotide preference of the WRKY54 DBD. (**B**) P, positive control; N, maltose-binding protein (MBP) only; W4, the W4 box shown in Figure 3A. The fEMSA banding shift patterns observed when combining WRKY54 DBD protein with fluorescence-labeled W4 and W4-like probes. (**C**) Plotted curves for the binding assays of the WRKY54 DBD protein with W4 and the artificially truncated W4-like nucleotides.

**Figure 6 ijms-23-02895-f006:**
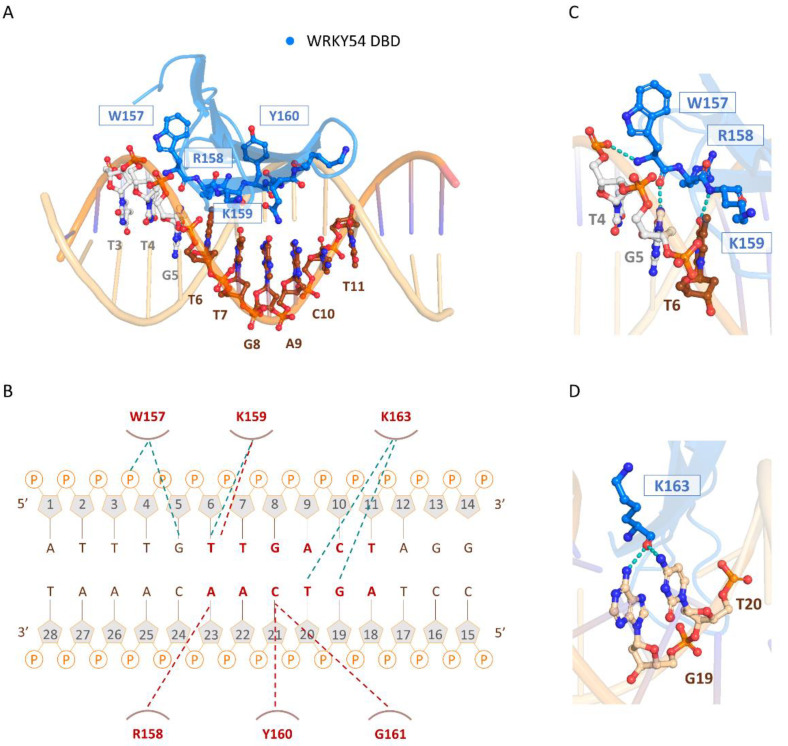
The binding mode of the WRKY54 DNA-binding domain (DBD) to W4 DNA. (**A**) Overall structure of the WRKY54 DBD and W4 complex. The nucleotides and amino acids involved in the protein–DNA interaction are displayed as a ball-and-stick model. (**B**) Summary of the interaction between the WRKY54 DBD and W4. The apolar interactions are indicated by red lines and the hydrogen bonds are represented by green lines. (**C**,**D**) Close-up view of the W4–WRKY54 DBD interaction interface. Hydrogen bonds are indicated with green dotted lines.

**Figure 7 ijms-23-02895-f007:**
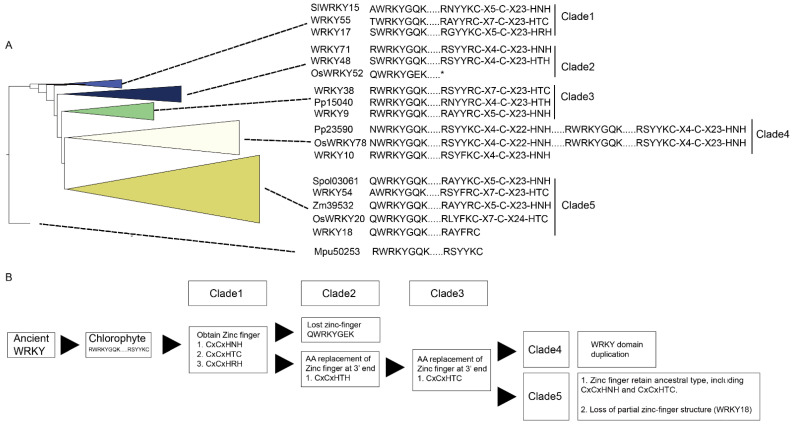
Evolutionary scenario for the WRKY domain. (**A**) Summary of the WRKY phylogeny constructed in this study. Representative WRKY domains from each clade are shown on the right. (**B**) Our proposed WRKY domain evolution process. The key zinc-finger amino acids in each clade are labelled. X denotes any unspecified amino acid. * denotes lack of zinc-finger motif.

**Table 1 ijms-23-02895-t001:** Identified WRKYs from selected species.

Species	No. of WRKY Gene	Taxanomy
*Porphyra umbilicalis*	x	Rhodophyta
*Micromonas pusilla*	2	Chlorophyte
*Physcomitrium patens*	21	Embryophyte
*Marchantia polymorpha*	12	Embryophyte
*Selaginella moellendorffii*	7	Tracheophyte
*Zea mays*	62	monocot
*Oryza sativa*	75	monocot
*Musa acuminata*	45	monocot
*Amborella trichopoda*	20	Angiosperm
*Aquilegia coerulea*	6	Eudicot
*Spinacia oleracea*	21	Pentapetalae
*Solanum lycopersicum*	39	Asterids
*Helianthus annuus*	46	Asterids
*Lotus japonica*	30	Fabidae
*Fragaria* x *ananassa*	54	Rosids
*Arabidopsis thaliana*	70	Rosids
*Prunus persica*	35	Rosids

**Table 2 ijms-23-02895-t002:** Dissociation rate constant (*K*_d_) values measured for W4– and W4-like–WRKY54 DBD pairs.

Probes	Sequence (5′ to 3′)	*R* ^2^	*K*_d_ (pM)
W4	ATTTG**TTGACT**AGG	0.97	163 ± 10.06
T13	ATTTG**TTGACT**AG	0.96	192 ± 22.5
T12	ATTTG**TTGACT**A	0.95	2195 ± 442.3
T12a	TTG**TTGACT**AGG	0.99	47 ± 6.2
T10	G**TTGACT**AGG	–	N.D.
T8	G**TTGACT**A	–	N.D.
T6	**TTGACT**	–	N.D.

N.D., nondetectable.

## Data Availability

Data supporting the reported results can be found in the Appendix A.

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
