# Peer review of "Insight into the Phylogeny and Binding Ability of WRKY Transcription Factors"

_ijms, 2022, doi:10.3390/ijms23052895_

Round 1
Reviewer 1 Report
The phylogenetic tree should be rebuilt with better, representative plant species included.
The manuscript entitled Insight into the phylogeny and binding ability of WRKY transcription factors presented by authors Hsin et al., seeks to gain insights into the evolutionary history and putative binding mechanism of WRKY TFs. This manuscript is well-written, and presents a unique evolutionary view on the WRKY TFs based on the appearance of main domains and motifs in various species in plant kingdom. Then, the authors find WRKY54, taking as an example, that binds to the W4 box of upstream region of SAG12.
However, one major concern should be addressed.
1) The authors retrieved WRKY sequences from 15 species. However, these 15 species can not represent the diversity of plant kingdom. None of the above species is tetraploid, hexaploidy, or other multi-ploidy. Plant species with different ploidy should be included, and allote- and autoploid species should also be considered.
Some minor comments can be found below.
Line 82: spell out “AtSIRK”
Line 92: state how many species
Line 254: Why is OsWRKY52 grouped in Clade 1? Explain it a bit in discussion
Line 259: Why is Mp7s001 grouped in Clade 2? Explain it a bit in discussion
Line 394: Suggest change “…and the abiotic stress response” to “…abiotic stress responses”
Line 397: Phrase “…previous studies have been unable…” as “…previous studies were unable… ”
Author Response
The phylogenetic tree should be rebuilt with better, representative plant species included.
The manuscript entitled Insight into the phylogeny and binding ability of WRKY transcription factors presented by authors Hsin et al., seeks to gain insights into the evolutionary history and putative binding mechanism of WRKY TFs. This manuscript is well-written, and presents a unique evolutionary view on the WRKY TFs based on the appearance of main domains and motifs in various species in plant kingdom. Then, the authors find WRKY54, taking as an example, that binds to the W4 box of upstream region of SAG12.
However, one major concern should be addressed.
1) The authors retrieved WRKY sequences from 15 species. However, these 15 species can not represent the diversity of plant kingdom. None of the above species is tetraploid, hexaploidy, or other multi-ploidy. Plant species with different ploidy should be included, and allote- and autoploid species should also be considered.
Reply: First, we thank for reviewer’s comments. Those species chosen in this study ranging from Chlorophyte, Embryophyte, Tracheophyte, monocot, basal angiosperm, Pentapetalae, Asterids and Rosids (details See Table 1). Therefore, main plant lineages are included. We accept the comments and choose Musa acuminate (triploid) and Fragaria x ananassa (octoploid) as our multi ploidy samples to retrieve their WRKY TFs. We obtain 45 WRKY TFs from Musa acuminate and 54 WRKY TFs from Fragaria x ananassa respectively. Updated WRKY TFs phylogeny shows 5 independent clades, consistent with our previous phylogeny. After checking Musa acuminate and Fragaria x ananassa WRKY sequence dataset, we find 7 and 12 two-WRKY domain containing WRKY TFs from Musa acuminate and Fragaria x ananassa respectively. In the updated phylogeny (Figure 1), two-WRKY domain containing WRKY TFs from Musa acuminate and Fragaria x ananassa grouped with other two-WRKY domain containing WRKY TFs used. Details are updated in the sections of results and discussion (line 254-272, line 279-285, line 429-449). By adding polyploidy species WRKY TFs helps to elucidate WRKT TFs evolution in plants.
Some minor comments can be found below.
Line 82: spell out “AtSIRK”
Reply: For clarity, we accept comment and spelled out AtSIRK (Arabidopsis thaliana senescence-induced receptor-like kinase) in line 82-83 in “with track change file”.
Line 92: state how many species
Reply: In Wang et. al, (2011) study, they aimed to solve WRKY TFs relationship of Arabidopsis thaliana. Therefore, only A. thaliana is used to identify relationship of its WRTY TFs. For clarity, we add species scientific name “A. thaliana” in line 90-91.
Line 254: Why is OsWRKY52 grouped in Clade 1? Explain it a bit in discussion
Reply: OsWRKY52 is now grouped into Clade 2 due to its WRKY motif similarity (Line 438-439).
Line 259: Why is Mp7s001 grouped in Clade 2? Explain it a bit in discussion
Reply: Mp7s001 is now grouped into Clade 5. This could due to its WRKY motif and zinc-finger motif similarity among Clade 5 WRKY TFs. We remove this description after checking updated phylogeny.
Line 394: Suggest change “…and the abiotic stress response” to “…abiotic stress responses”
Reply: Following suggestion, we changed “…and the abiotic stress response” to “…abiotic stress responses” in line 405 and line 406.
Line 397: Phrase “…previous studies have been unable…” as “…previous studies were unable…
Reply: Following suggestion, we changed “…previous studies have been unable…” to “…previous studies were unable… in line 408.
Reviewer 2 Report
The manuscript “Insight into the phylogeny and binding ability of WRKY transcription factors” describes the phylogeny and adeptness of binding based on W-box sequences. The authors used ample numbers of WRKY genes to figure out the origin of the plant kingdom. They also provided insights on the protein-DNA recognition mechanism and the sequence-selective binding of the highly conserved WRKY TFs. This research work is a useful addition to the phylogeny of WRKY and their molecular binding mechanism. It should be recognized and accepted for publication in IJMS. However, some of the minor issues should be addressed to improve the manuscript mentioned given below.
Please revise and improve the manuscript for the English language quality. The grammar of the manuscript needs to be improved. One example is in the first sentence of the abstract which is carrying unnecessary and excessive commas. Please scan the whole manuscript with those kinds of errors. One more example is in line 35 where "The” is missing before group 1. Furthermore, be consistent in writing style either passive or active. Half of the material and method section is written in passive and half in active. So is true for results and discussion.
Please be consistent in italicizing genes names. Two examples are in lines 65 and 135. Please check throughout the manuscript.
Please be consistent when writing a full form and abbreviations. The authors used full form A. thaliana in the introduction but used abbreviation in line 112.
What is the reason to use the selective species to use as a template? Please give a short reason either in the introduction, methods part, or discussion.
Please be consistent in reference style. Line 190
In a discussion section, the authors again explained their own results instead of giving relevant comparative or supportive information from other research works. Line 417 to 442 also line 448 to 475. I can see only the elaboration of the results instead of a discussion about the findings. The discussion part is too long it should be more precise.
It would be better to add a separate heading "conclusions” and state a clear take-home message for the readers and researchers.
The author’s contribution part is missing.
Please carefully revise the references. The reference style is not according to the journal. Several genes names should be italic.
Author Response
Please revise and improve the manuscript for the English language quality. The grammar of the manuscript needs to be improved.
One example is in the first sentence of the abstract which is carrying unnecessary and excessive commas. Please scan the whole manuscript with those kinds of errors.
Reply: First, we thank for your comments. For clarity, we rewrite this sentence in abstract in line 13-14 in “with track change file”. The manuscript has been checked by English Editing.
One more example is in line 35 where "The” is missing before group 1.
Reply: Following suggestion, we add “The” before group 1 in line 36.
Furthermore, be consistent in writing style either passive or active. Half of the material and method section is written in passive and half in active. So is true for results and discussion.
Please be consistent in italicizing genes names. Two examples are in lines 65 and 135. Please check throughout the manuscript.
Reply: Thanks for reminding. We have carefully checked the manuscript and rewrite the writing style in almost passive form (line124-125, 179, 191, 193-194, 204-206, 218, 227-228, 235, 239-240, 252-253, 304-405, 347, 364, 373, 378-379, 424-425, 435, 496-498, 515, 551). In addition, we refer to gene in the manuscript, then the gene writing style will be italic. We correct “WRKY53” to “WRKY53” in line 65. We also correct the writing style in line 67-70 to show these are referring to gene instead of protein.
Reply: In line 139 and 140, we refer WRKY54 gene. So, we retain original italic writing style.
Please be consistent when writing a full form and abbreviations. The authors used full form A. thaliana in the introduction but used abbreviation in line 112.
Reply: For consistency, we now use full form of A. thaliana in line 113.
What is the reason to use the selective species to use as a template? Please give a short reason either in the introduction, methods part, or discussion.
Reply: These species are chosen because they represent main classification of plants. For example, Micromonas pusilla represents Chlorophyte. We provide our reason in “materials and methods” section (Line 115-120).
Please be consistent in reference style. Line 190
Reply: Thanks for reminding. We upgrade the reference format in Line 194.
In a discussion section, the authors again explained their own results instead of giving relevant comparative or supportive information from other research works. Line 417 to 442 also line 448 to 475. I can see only the elaboration of the results instead of a discussion about the findings. The discussion part is too long it should be more precise.
Reply: Thanks for reminding. After reading, we rewrite aforementioned paragraphs (line429-line475).
Reply: To show concise, we remove first paragraph of section “3.2 Molecular binding mechanism for specific WRKY–W-box interaction” (from line 482 to 488).
It would be better to add a separate heading "conclusions” and state a clear take-home message for the readers and researchers.
Reply: Following suggestion, we now add “Conclusion” section from line 562-569.
The author’s contribution part is missing.
Reply: Thanks for reminding, we add author’s contribution section from line 575-578.
Please carefully revise the references. The reference style is not according to the journal. Several genes names should be italic.
Reply: Thanks for reminding. We update reference style of genes names and species scientific names in our reference list (line 588; line 596; line 651). In addition, reference format is unified (Line 580-663).
Round 2
Reviewer 1 Report
Modifications addressed my comments